# Gut Microbiota and Its Metabolites in Atherosclerosis Development

**DOI:** 10.3390/molecules25030594

**Published:** 2020-01-29

**Authors:** Magdalena D. Pieczynska, Yang Yang, S. Petrykowski, Olaf K. Horbanczuk, Atanas G. Atanasov, Jaroslaw O. Horbanczuk

**Affiliations:** 1Department of Molecular Biology, Institute of Genetics and Animal Breeding of the Polish Academy of Sciences, Postepu 36A Street, 05-552 Jastrzebiec, Poland; y.yang@ighz.pl (Y.Y.); s.petrykowski@ighz.pl (S.P.); atanas.atanasov@univie.ac.at (A.G.A.); 2Institute of Genetics and Biotechnology, Faculty of Biology, University of Warsaw, Pawinskiego 5A Street, 02-106 Warsaw, Poland; 3Institute of Clinical Chemistry, University Hospital Zurich, Wagistrasse 14, 8952 Schlieren, Switzerland; 4Institute of Human Nutrition Sciences, Warsaw University of Life Sciences (WULS-SGGW), 159c Nowoursynowska, 02-776 Warsaw, Poland; olaf_horbanczuk@sggw.pl; 5Department of Pharmacognosy, University of Vienna, 1090 Vienna, Austria; 6Institute of Neurobiology, Bulgarian Academy of Sciences, 23 Acad. G. Bonchev str., 1113 Sofia, Bulgaria; 7Ludwig Boltzmann Institute for Digital Health and Patient Safety, Medical University of Vienna, Spitalgasse 23, 1090 Vienna, Austria

**Keywords:** atherosclerosis, bioactive compounds, gut microbiota, metabolites, natural products

## Abstract

Gut microbiota metabolites have a great influence on host digestive function and body health itself. The effects of intestinal microbes on the host metabolism and nutrients absorption are mainly due to regulatory mechanisms related to serotonin, cytokines, and metabolites. Multiple studies have repeatedly reported that the gut microbiota plays a fundamental role in the absorption of bioactive compounds by converting dietary polyphenols into absorbable bioactive substances. Moreover, some intestinal metabolites derived from natural polyphenol products have more biological activities than their own fundamental biological functions. Bioactive like polyphenolic compounds, prebiotics and probiotics are the best known dietary strategies for regulating the composition of gut microbial populations or metabolic/immunological activities, which are called “three “p” for gut health”. Intestinal microbial metabolites have an indirect effect on atherosclerosis, by regulating lipid metabolism and inflammation. It has been found that the diversity of intestinal microbiota negatively correlates with the development of atherosclerosis. The fewer the variation and number of microbial species in the gut, the higher the risk of developing atherosclerosis. Therefore, the atherosclerosis can be prevented and treated from the perspective of improving the number and variability of gut microbiota. In here, we summarize the effects of gut metabolites of natural products on the pathological process of the atherosclerosis, since gut intestinal metabolites not only have an indirect effect on macrophage foaming in the vessel wall, but also have a direct effect on vascular endothelial cells.

## 1. Introduction

The past years have attracted remarkable attention to gut and its accompanying microbial communities, and how their diversity influence the biology of many disorders and diseases [1,2,3,4,5,6]. Many studies have proven that the composition of intestinal microbial populations are related to the physiology of the host and the occurrence of several diseases [7,8,9,10]. Gut microbiota metabolites have a great influence on host digestive function and body health itself [11]. The effects of intestinal microbes on the host metabolism and nutrients absorption are mainly due to regulatory mechanisms related to serotonin, cytokines, and metabolites. Gut microbes decompose the food and drugs consumed by the host, resulting in the production and secretion to the circulatory system of small-molecule secondary metabolites, which exert additional effects when reaching various parts of the host [7,12]. Therefore, microbial metabolites can promote the secretion of cytokines via immune cells of the host [13]. Occasionally, the concentration of secondary metabolites in the blood can reach or exceed the level of the drug dose, suggesting that the metabolism of intestine microbes can be considered as so-called pharmaceutical process of the host itself. Moreover, several beneficial bioactive like polyphenolic compounds (protocatechuic acid, quercetin-3-glucuronide, α-asarone, gallic acid, equol, enterolactone, enterodiol and urolithins) can be only absorbed by the host after being decomposed by intestine microbes [11,14,15]. Therefore, the metabolism of gut microbes has a potential regulatory effect on the host health, and the prevention, occurrence and treatment of atherosclerosis and subsequent cardiovascular diseases (CVDs). Recent studies have proven that intestinal microbes are a major cause of atherosclerosis, related to the substance called trimethylamine oxide (TMAO) [16].

This review discusses the effect of gut microbiota metabolism on the atherosclerosis development, emphasizing macrophage cholesterol efflux, exploring its application in health monitoring, disease prevention and possible treatment.

## 2. Gut microbiota Metabolism of Nutrients and Other Food Components

In ecological niche of any living body, host and a large number of microbial populations have co-adapted together. Through the long-term co-evolution with their hosts, gut microbial species have become the important “functional organs” that play a substantial role in maintaining host health [2]. Intestinal microorganisms participate in the regulation of multiple metabolic pathways, forming the immune-inflammatory axis, that are linked to gut, liver and brain functions [17]. Such microbial populations participate in the metabolism of digestive system. Additionally, they are capable of producing a variety of secondary metabolites that accumulate in the host blood, contributing to the systemic effect on the host [18].

Hosts and their accompanying gut microbiota produce a large number of small molecular metabolites during co-metabolism of food and/or exogenous substances [19]. Part of these metabolites play a pivotal role in the transmission of host cells to the intestinal environment. Different segments of intestine are harbored by different species of microbiota, and each part has the specific effect on chemical signals of host-microbiota information exchange [20]. Such potential ‘’signal molecules’’ include low-molecular weight metabolites, polypeptides, proteins, etc. They possess an ability to replace certain indirect immune regulatory pathways. Currently, more than 300 microbial-host metabolites have been identified through non-targeted and targeted metabolomics analysis, including bile acids, short-chain fatty acids, amino acids, benzoyl and phenyl derivatives, hydrazine derivatives, lipids, choline, phenols, vitamins, hormones and polyamines [21]. Multiple studies have repeatedly reported that the gut microbiota play a fundamental role in the absorption of bioactive compounds [22].

## 3. Gut microbiota in Atherosclerosis

Pervious comparative study based on pyrosequencing of microbes derived from plaques and intestinal samples from atherosclerotic patients has shown that the microorganisms in the plaque are consistent with some of the species in intestine, suggesting that microbial species might have their root in the gut [4]. Another study undergone on atherosclerotic patients with present clinical symptoms has proven that their gut microbes are rich in collins, which easy synthesize peptidoglycan for activating neutrophils and promoting the release of pro-inflammatory factors [23]. Additionally, scientists have previously demonstrated that patients with heart failure possess gut microbiota populations dominated by large number of Gram-negative bacteria and *Candida* fungi species, that are far exceeding the incidence of healthy individuals. Such an overgrown gut microbiota causes an increase in intestinal permeability and promote the progression of heart failure [24]. This data indicates that intestinal microbes may promote or accelerate the occurrence of atherosclerosis and other CVDs by transferring and changing the composition of microbes related to disease, so-called dysbiosis. In addition to the alterations in microflora composition, the metabolic potential of gut microbiota has been referred to as a key factor in the development of atherosclerosis.

Atherosclerosis, the major cause of CVDs is a chronic inflammatory disease caused by the formation of plaques composed of necrotic cores, calcified regions, accumulated modified lipids, inflamed smooth muscle cells, endothelial cells, leukocytes, and foam cells. Atherosclerosis is strongly characterized by impaired endothelial function and lipid metabolism [25], whereas its inflammation with plaque development. Endothelial dysfunction enables entry of lipoproteins and migration of inflammatory cells into the initial layer of the artery [26]. Consequently, multiple cells are activated inside the plaque. Cholesterol crystals trigger activation of inflammasome and stimulate release of inflammatory cytokines. Accumulation of large amounts of lipids in the immune cells results in the cell death, resulting in the initiation of necrosis [27]. In turn, the superior plaques are formulated due to necrotic core maintenance. Furthermore, tertiary lymphoid organs are developed in the adventitial layer of the vessel wall, providing inflammatory cells and mediators into the plague, causing advance plaque inflammation. Neutrophils, mast cells and innate lymphoid cells, such as natural killers (NKs) play a pivotal role in the development of atherosclerotic inflammation [28]. Moreover, macrophages and dendritic cells are also involved in atherosclerosis induction and progression.

The risk factors for inducing atherosclerosis include hypertension, abnormal cholesterol levels, smoking, diabetes and obesity. In addition, recent studies have found that intestinal microbes are a major cause of atherosclerosis, and the effect of intestinal microbes on atherosclerosis is related to TMAO [29].

In 2015, researchers from the Cleveland Clinic reported that lecithin and L-carnitine (present in red meat, egg yolk, etc.) occurring in the diet can be converted into TMAO under the action of intestinal microorganisms, which in turn promote the development of atherosclerosis and accelerate the pathological process of cerebrovascular diseases [30]. This group found that intestinal microbes play an important role in the formation of TMAO. H3 labeled lecithin used for feeding the mice was observed to have a large amount of H3-labeled TMAO in their blood. However, another study demonstrated that administered broad-spectrum antibacterial drugs to animals before conducting the experiments were able to show inhibition of intestinal microbes, and consequently diminish the TMAO in the blood to the level lower that the one of control individuals [31]. In sum, foods rich in lecithin, choline, and carnitine can be metabolized by intestinal microbes to form trimethylamine (TMA, a colorless gas with a foul odor), which can enter the liver and be oxidized by the flavin monooxygenase (FMO) to TMAO (especially the highest activity FMO3). The elevated TMAO in the blood holds the explanation for the incidence of atherosclerosis [32]. Therefore, the level of TMAO in the blood is related to the diet, intestinal microbes, FMO3 activity, gender, and heredity of host [33]. The mechanism by which TMAO promotes atherosclerosis may have the following aspects: (1) inhibits cholesterol reverse transcription [32]; (2) up-regulates the expression of macrophages CD36 and scavenger receptor A1 (SR-A1), promoting cell uptake, accumulating cholesterol and promoting the formation of foam cells; (3) down-regulates the expression of cholesterol absorption targets ABCG5/8 and NPC1L1 to lower the intestinal tract of cholesterol, which could affect cholesterol metabolism; (4) reduces the expression of cytochrome P450 (CYP) 7A1 and 27A1 in liver, thereby reducing the bile acid transport and cholesterol clearance and promotes vascular inflammation by activating monocytes through mitogen-activated kinase and nucleic acid factor-κB signaling pathway [33] (Figure 1). Therefore, it can be strongly concluded that intestinal microbial metabolites have an indirect effect on atherosclerosis, by regulating lipid metabolism and inflammation. Therefore, atherosclerosis can be prevented and treated from the perspective of improving the number and variability of gut microbiota. Furthermore, scientists have already found that the diversity of intestinal microbiota negatively correlates with development of female atherosclerosis. The fewer the variation and number of microbial species in the gut, the higher the risk of developing atherosclerosis [34]. In 2016, a research group discovered a substance called resveratrol that can reduce the risk of atherosclerosis by regulating the quantity and status of intestinal microbial population [16]. They also identified that treatment targeted at gut microbiota and their metabolites can improve arterial aging. Moreover, resveratrol has repeatedly proven health-promoting benefits, including anti-cancer and chemopreventive effects [15,35] with potential benefits for the atherosclerosis and other CVDs [36]. Therefore, the exploration of natural products with their bioactive like polyphenolic compounds which use gut microbes as potential therapeutic targets has a gradually growing attention.

Gut microbiome composition appears to be less fermentative and more inflammatory in atherosclerotic patients. The observed alterations within the atherosclerotic gut microbiota composition include higher abundance of *Escherichia coli*, *Klebsiella* spp., *Enterobacter aerogenes*, *Streptococcus* spp., *Lactobacillus salivarius*, *Solobacterium moorei*, and *Atopobium parvulum,* whereas depleted abundance of gut species such of *Bacteroides* spp., *Prevotella copri*, and *Alistipes shahii* [7]. Additionally, bacteriophages are also more diverse during atherosclerosis onset. Interestingly, previous randomized double-blind clinical trial [37] has shown that administrating probiotic of *Lactobacillus plantarum* has the ability to reduce circulating cholesterol levels and therefore restrain the formulation of atherosclerotic plaques in hyper-cholesterol patients.

## 4. The role of Macrophages in Atherosclerosis and Reverse Cholesterol Transport

In the early stage of atherosclerosis, monocytes gradually adhere to the vessel wall in the circulatory system and invade the vascular endothelium under the action of chemotactic molecules, which are produced by local inflammation and differentiate into macrophages [38]. Such a process is regulated by scavenger receptors. Furthermore, in the localized part of atherosclerosis, scavenger receptors phagocytose cholesterol through the recognition of oxLDL-mediated macrophage sources, resulting in a large accumulation of lipids in macrophages and the formation of lipid streaks on the vessel wall. The accumulation of a large amount of lipids such as cholesterol esters, makes the space occupied by intracellular subcellular organelles smaller, causing them to gradually degenerate or in some situations fully disappear, term so-called macrophage foaming. The fatty streak represents the earliest visible damage of atherosclerosis and mainly consists of macrophage-derived foam cells. In the presence of risk factors including hypertension, hyperlipidemia and hyperglycemia, the fatty streaks can progress faster to more complex lesions. In summary, the accumulation of lipid-laden macrophages in the subendothelial or neointima constitutes form the first step of atherogenesis [39].

Reverse cholesterol transport (RCT) is a process in which cholesterol is transported from the surrounding tissues to the liver and then excreted in the form of circulation or bile acid. It is an important physiological pathway for the body to remove excess cholesterol, which may represent a novel strategy to reduce atherosclerotic plaque burden and subsequent CVDs. This represents one of the most important step is high-density lipoprotein (HDL)-mediated cholesterol efflux in macrophages, where the lipoproteins remove excess cholesterol from cells [40]. Conversely, if a natural product regulated by macrophage cholesterol efflux and cholesterol distribution, which promote RCT indirectly, it could play a potential role in cardiovascular protection by reducing the lipid deposition in the arterial intima, and in turn slowing down the pathological processes of atherosclerosis.

## 5. The Role of Potential Gut Metabolites Derived from Bioactive like Polyphenolic Compounds and Prebiotics on Atherosclerosis

Bioactive like polyphenolic compounds, prebiotics and probiotics are the best known dietary strategies for regulating the composition of gut microbial populations or metabolic/immunological activities, which are called “Three “p” for gut health” [18]. Polyphenols are a class of plant secondary metabolites which have been found to possess various antioxidant, anti-inflammatory or anticarcinogenic biological activities (Figure 2). It is known that many natural products contain a large amount of polyphenols that possess the functions of protecting cardiovascular system and reducing inflammatory responses [41,42]. Although, polyphenols are thought to prevent development of chronic diet-related diseases, most of them could not be absorbed directly by the small intestine. Therefore, their bioavailability and impact on the host mostly depends on the function of the gut microbiota and their component’s conversion [14] (Table 1). Intestinal microbes play a crucial role in converting dietary polyphenols into absorbable bioactive substances. Moreover, some intestinal metabolites derived from natural polyphenol products have more biological activities than their own fundamental biological functions.

Prebiotics are a special type of food materials, which can promote proliferation and activity of beneficial gut probiotics, thereby improving the health of the host [65]. It mainly includes various oligosaccharides (composed of two to 10 molecular monosaccharides), which are functional oligosaccharides. Many studies have shown that changes in gut microbiota composition, activity and metabolism are the result of probiotic supplementation towards many diseases, such as obesity, atherosclerosis, and more metabolic disease [66]. Therefore, the beneficial effects of prebiotics are also inseparable from intestinal microbes. Human intestinal tract is not only the main place for digestion and drug absorption, but also the important home for regulating the physiological functions and assisting in the treatment of diseases through the metabolism of diet modulated by intestinal microbes.

A variety of metabolites are produced and secreted from gut microbiota including amines methylamines, polyamines, short-chain fatty acids (SCFAs), TMAO and secondary bile acids (BAs) that contribute to atherosclerotic onset. Formation of TMAO play a pivotal function in the pathogenesis of atherosclerosis [37]. The higher amount of plasma TMAO, the higher incidence of atherosclerotic onset and formation of the atherosclerotic plaque surface [31]. Previous clinical study on healthy and affected by chronic heart failure individuals has proven that plasma quantity of TMAO has a direct effect on atherosclerosis development [67]. Furthermore, TMAO can lead to atherosclerosis by suppressing reverse cholesterol transport and modulating the activity of cholesterol transporters in macrophages. Moreover, a number of studies have shown that SCFAs take part in all stages of atherosclerosis development [68].

In here, we summarize part of recent research on association between gut metabolites of natural products and atherosclerosis, based on the physiological functions of gut microbiota. More specifically, the focus on the effects of gut metabolites of natural products on the early pathological process of atherosclerosis. These gut intestinal metabolites not only have an indirect effect on macrophage foaming in the vessel wall, but also have a direct effect on vascular endothelial cells.

### 5.1. Protocatechuic Acid

Protocatechuic acid (PCA) is a bioactive compound present in some medicinal herbs used in natural medicine. It is also a very common compound present in various fruits, such as plums [43], gooseberries, grapes [44] and almond ordinate nuts [45]. Protocatechuic acid is also found in several plants and spices, including star anise, melissa, rosemary and cinnamon [69].

It is a major metabolite of fruits/vegetables-derived anthocyanins (such as cyanidin-3-*O*-β-glucoside) generated by gut microbiota [38,46]. Interestingly, it has been previously shown that gut microbiota metabolites of natural product display a promising antiatherogenic effect at physiologically reachable concentrations. They can accelerate the cholesterol efflux in AcLDL-loaded MPMs or THP-1 macrophages correlated with the regulation of miRNA-10b-ABCA1/ABCG1 cascade [38]. In another way, PCA has been shown to inhibit ICAM1 and vascular cell adhesion molecule 1 (VCAM-1)-dependent monocyte adhesion to activated HUVEC endothelium, as well as CCL2-mediated monocyte transmigration, thereby reducing the development of atherosclerosis in ApoE−/− mice [46]. It also exhibits inhibitory effects on VSMC proliferation (induced by oleic acid) by activating AMPK and arresting cell cycle at G0/G1 phase in A7r5 smooth muscle cell line [47]. Therefore, it could be also explored as potential novel molecule for atherosclerotic prevention and treatment.

### 5.2. Quercetin-3-glucuronide

Quercetin-3-glucuronide is the most antioxidative metabolite from quercetin natural flavonoid family, commonly identified in almost every vegetable and fruit, including red onion, cranberry, blueberry and fig [48]. It has been reported to inhibit the formation of macrophage foam cells by suppressing the expression of scavenger receptor class A type 1 (SR-A1) and CD36 in RAW 264.7 cells. Guercetin glucuronide is specifically accumulated in atherosclerotic lesions in human aorta, especially in the macrophage-derived foam cells [49]. It has been shown that the metabolites of dietary quercetin might exhibit potential anti-atherosclerotic effects in injured/inflamed arteries with activated macrophages. In another way, in PC12 cells, Q3GA could significantly suppress the formation of reactive oxygen species (ROS), which are mainly produced by vascular cells that are implicated as possible risk factors in the progression of atherosclerosis [50]. These results show a novel approach for the prevention of atherosclerosis and many other CVDs by flavonoid diet.

### 5.3. 2,4,5-Trimethoxycinnamic Acid

α-Asarone is biologically active compound of the phenylpropanoid class found in certain plants of *Acorus* and *Asarum* origin [51]. It has proven hypocholesterolemic effects [70]. 2,4,5-Trimethoxycinnamic acid, the major and non-toxic metabolite of α-asarone represents its major pharmacological properties like regulating effects on blood lipid, lowering the blood level of total cholesterol, high-density lipoprotein cholesterol and low-density lipoprotein cholesterol in hypercholesterolemic rats [52,53]. Therefore, this compound is recommended for further investigation as a potential hypocholesterolemic and cardiovascular protective factor.

### 5.4. Gallic Acid

Gallic acid (GA), which represents anthocyanin metabolites can be commonly identified in gallnuts, sumac, witch hazel, tea leaves and oak bark [11,54]. Esters of gallic acid have found multiple approaches in food, cosmetic, and the pharmaceutical industry. Moreover, gallic acid is used in tanning, ink dyes, and the manufacture of paper. Gallic acid and its derivatives have proven to exert diverse potential health beneficial features. Particularly, It has been found to improve atherosclerosis through a vasorelaxant and antihypertensive effect. It has been shown to increase nitric oxide (NO) levels, by increasing phosphorylation of endothelial nitric oxide synthase (eNOS) in RAW 264. 7 macrophage [55]. Furthermore, it has been indicated that inhibited angiotensin-I converting enzyme (ACE) causes blood pressure reduction in spontaneously hypertensive rats comparable to captopril [71]. These results suggest that GA display multiple therapeutic properties and has a great potential for atherosclerosis prevention.

### 5.5. Equol

Equol, next to genistein, daidzein, and ipriflavone belong to the category of isoflavones. It can be derived from soybeans and other plant sources [56,57]. Equol is a chiral molecule produced by certain intestinal bacteria through the metabolism of daidzein. However, the specific bacterial species of intestinal microflora involved in the production of equol has not been discovered yet. Two isomers, *S* and *R*, have been identified until now. Currently, most common metabolites found in the body are *S*-type. Their biological activities are based on anti-oxidation, dual activities of estrogen and anti-estrogen, dual activities of proliferation and anti-proliferation. They have been reported to protect against atherosclerosis, CVDs and tumors [72]. It has been implicated that equol could improve the atherosclerosis by attenuation of endoplasmic reticulum stress, and partially, by activating the Nrf2 signaling pathway [21]. Previous clinical trial has shown that equol may display a fundamental atheroprotective properties of isoflavones in Japanese men [73]. The evidence from another observational studies and short-term randomized controlled trials has suggested that equol is anti-atherogenic and improves arterial stiffness and may prevent coronary heart disease [58]. Therefore, equol, a metabolite of daidzein gut microbes, has a significant developmental prospect for cardiovascular health and disease prevention.

### 5.6. Enterolactone and Enterodiol

Enterodiol and enterolactone are main gut metabolites of secoisolariciresinol diglucoside, an antioxidant isolated from flaxseed, which is one of the richest sources of lignans [74]. Products characterized by high fiber content such as vegetables, fruits and berries represent high lignans and enterolactone sources, as they bind to the fiber component [75]. Both of these metabolites have obvious antioxidant activity and affect a number of atherosclerosis-relevant proteins. Such lignan metabolites have been established as potential biomarkers of vascular health, suggesting a protective role in hypercholesterolemic atherosclerosis [59,60].

### 5.7. Urolithins

Urolithins are ellagitannin-derived gut microbiota metabolites in human, which are divided into two types: A and B. They are produced by gut microbiota and are natural metabolites of Ellagitannins, a class of bioactive compounds found in pomegranates, strawberries, walnuts and other fruits and nuts [61]. Urolithins have been reported to have important effects on atherosclerosis development, with the ability of monocytes to adhere to endothelial cells uptake and efflux of cholesterol by macrophages [76].

On one hand, a recent study has indicated that urolithin A show anti-atherosclerotic activity in Wistar rat through activation of class B scavenger receptor and Nef2 signaling pathway [62]. On the other hand, it has been demonstrated that Urolithin B decrease lipid plaque deposition in apoE-/- Mice, which in turn increase the early stages of RCT in oxidized low-density lipoprotein (ox-LDL) treated macrophages cells [44]. Therefore, these urolithins may represent the basis for exploration of novel bio-treatment for atherosclerosis.

### 5.8. Others

Most of the intestinal microbial metabolites derived from natural products have a variety of biological activities, whether it is primary, secondary or even a tertiary metabolite. They are able to directly affect the development of atherosclerosis through the regulation of macrophages and blood lipids, and also indirectly through the effects on vascular endothelial cells [37].

It has been determined that proanthocyanidins are extensively metabolized by gut microbiota. As a major microbial metabolite of proanthocyanidin, 5-(3’,4’-dihydroxyphenyl-γ-valerolactone) prevents THP-1 monocyte-endothelial cell adhesion by downregulating expressions of vascular cell adhesion molecule-1 and monocyte chemotactic protein-1, which are biomarkers of atherosclerosis [77]. This tertiary metabolite has the same potential in prevention of atherosclerosis as its original and former state. Former work has demonstrated three gut metabolites of dietary phosphatidylcholine - choline, betaine, and TMAO. These phospholipid metabolites are becoming the novel independent predictors for the risk of a clinical vascular event [31].

## 6. Conclusions

With the advances in modern technology and still growing interests in gut microbiota, it has been recognized that commensal microorganisms that habitat human body are not only passengers in their host, but can actually drive many fundamental functions. This review has highlighted atherosclerosis and some of CVDs areas in which gut microbiota and their metabolites are considered to play a key regulatory function in atherosclerosis formation through reverse cholesterol transport and accumulation of macrophage cholesterol, resulting in atherosclerotic plaques formation composed of lipids, cholesterol, calcium and other substances in blood vessels. We have highlighted specific treatments including usage of pro/prebiotics and bioactive like polyphenolic compounds (protocatechuic acid, quercetin-3-glucuronide, α-asarone, gallic acid, equol, enterolactone, enterodiol and urolithins) that can improve gut dysbiosis and prevent from atherosclerosis. Better understanding the mechanisms and contribution of gut commensal microbiota to the development of atherosclerosis and consequently group of CV diseases, might unable to develop new strategies to treat and prevent from diseases by regulating gut microbiota composition. In addition, in some cases, gut microbiome can be used to detect diseases associated with the gut prior to routine diagnosis. In the future, it might provide a chance to accurately stratify patients and create more effective treatments for atherosclerosis.

## Figures and Tables

**Figure 1 molecules-25-00594-f001:**
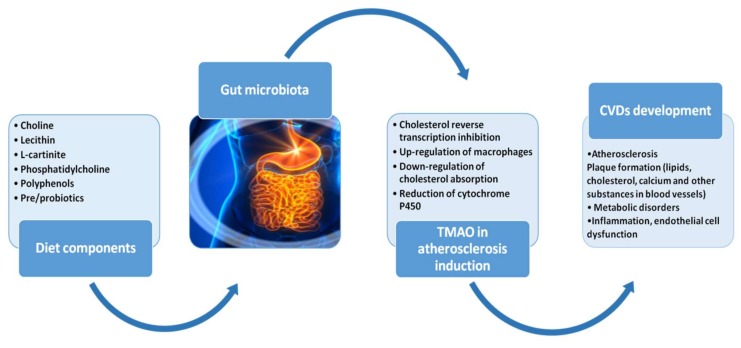
Figure displays the role of gut microbiota in development of atherosclerosis and subsequent CVDs. Dietary components can affect gut microbiota contributing to its dysbiosis. Lecithin, phosphatidylcholine, l-carnitine are metabolized into choline, which is converted to trimethylamine (TMA) by the gut microbiota. TMA is oxidized into trimethylamine N-oxide (TMAO) by hepatic flavin monooxygenases (FMO3). TMAO can accelerate atherosclerosis by inhibiting reverse cholesterol transport and accumulating macrophage cholesterol, resulting in atherosclerotic plaques formation composed of necrotic cores, calcified regions, accumulated modified lipids, inflamed smooth muscle cells, endothelial cells, leukocytes, and foam cells in blood vessels. Specific treatments including usage of pro/prebiotics and dietary polyphenols can improve gut dysbiosis and prevent from atherosclerosis.

**Figure 2 molecules-25-00594-f002:**
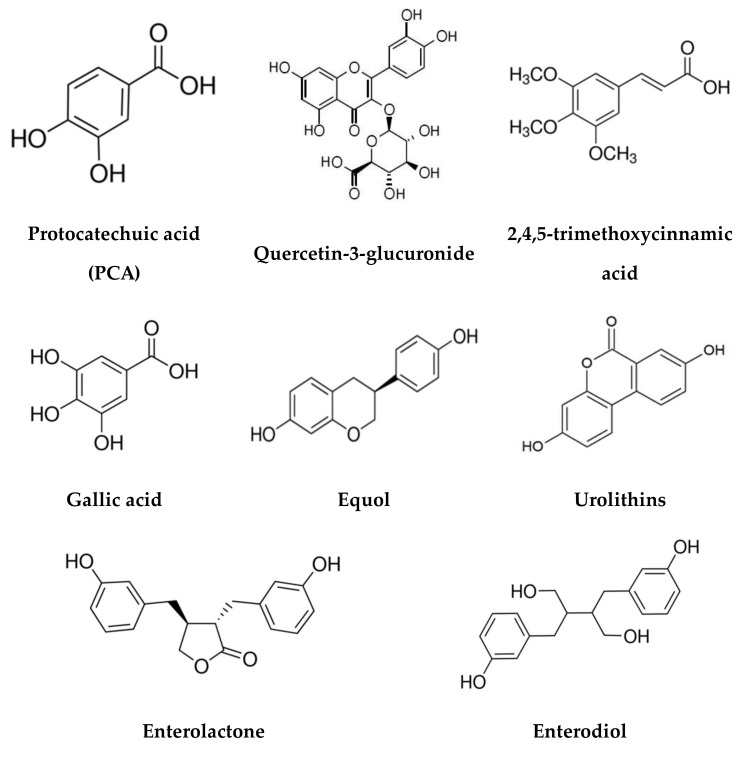
Chemical structures of bioactive like polyphenolic compounds.

**Table 1 molecules-25-00594-t001:** Natural products, their precursors and corresponding bioactive compounds, metabolites forms and gut enzymes for their production with potential beneficial effects on the pathological process of atherosclerosis and CVDs.

Compound Substrate Precursors	Bioactive like Polyphenolic Compounds	Gut Microbial Enzymes	Metabolites	Natural Sources	Potential Mechanism in Atherosclerosis and CVDs Prevention	References
Ferrulic acid (4-hydroxy-3- methoxycinnamic acid)	**Protocatechuic acid (PCA)**	Hydroxylase A Hydroxylase B	Sulfates Glucuronides	Plums Gooseberries Grapes Almonds Melissa Rosemary Cinnamon	Inhibits ICAM1 and vascular cell adhesion molecule 1 (VCAM-1) Inhibits CCL2-mediated monocyte transmigration Exhibits inhibitory effects on VSMC	[43,44,45,46,47]
Miquelianin	**Quercetin-3-glucuronide**	Hydrolase β-glucuronidase	Sulfates	Red onion Cranberry Blueberry Fig	Inhibits the formation of macrophage foam cells Suppresses the expression of scavenger receptor in cells proliferation	[48,49,50]
α-asarone	**2,4,5-trimethoxycinnamic acid**	Hydroxy-3- -Methylglutaryl Coenzyme A Reductase	Carboxylic acid	Acorus Asarum	Lowers blood lipid Lowers total cholesterol, high-density and low-density lipoprotein cholesterol	[51,52,53]
Gallotannin and Ellagitanni	**Gallic acid**	Tannin-acyl-hydrolase	Sulfates	Gallnuts Tea leaves Oak bark	Increases phosphorylation of endothelial nitric oxide synthase Inhibits angiotensin-I converting enzyme (ACE), leading to reduced blood pressure	[11,54,55]
Daidzein	**Equol**	Daidzein reductase	Enantiomer	Soybeans	Attenuates endoplasmic reticulum stress Activates the Nrf2 signaling pathway Improves arterial stiffness	[21,56,57,58]
Secoisolariciresinol, Matairesinol, Lariciresinol, Pinoresinol Sesamin	**Enterolactone and Enterodiol**	β-Glucuronidase Dehydroxylase Dehydrogenase	Eestradiols	Flaxseed Cereals Berries	Affects a number of atherosclerosis-relevant proteins Potential biomarkers of vascular health	[59,60]
Ellagitannins	**Urolithins**	Lactonase Decarboxylase Dehydroxylase	Glucuronides	Pomegranates, Strawberries Walnuts	Activates class B scavenger receptor and Nef2 signaling pathway Decreases lipid plaque deposition	[61,62,63,64]

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
