# Peer review of "Gut Microbiota and Its Metabolites in Atherosclerosis Development"

_molecules, 2020, doi:10.3390/molecules25030594_

Round 1

Reviewer 1 Report

Pieczynska et al described the potential effects of drugs. My only concern is to know the metabolic side effects of the drugs inside the gut. Also , kindly provide the structural mechanism of any of the reported drug for better understanding.

Author Response

"Please see the attachment ".

Reviewer 2 Report

Please see my revision in attach

thanks

Author Response

"Please see the attachment ".

Reviewer 3 Report

In this manuscript (ID#: molecules-678194 ), titled “gut microbiota and its metabolites in atherosclerosis development”, the authors, Pieczynska et al, reviewed the information regarding gut microbiota and its metabolites in the development of atherosclerosis. The role of gut microbiota and its metabolites in the cardiovascular diseases are the hot topic in recent research. It is a fast developing research area. Thus, the information that this manuscript discussed is important. However, there are several major concerns are needed to be addressed, which are listed in the following paragraphs:

There are so many reviews have been published in this topic. The special focus of this review has to be pointed out at the beginning of manuscript. The second item, “gut microbiota and their role in modulation of cardiovascular diseases”, is out of the scope of current review, which is focusing on atherosclerosis. This manuscript is discussing about gut microbiota in atherosclerosis. However, little information regarding the microbiota alterations associated with atherosclerosis was provided. Please provide more detail information about the alterations of gut microbiota during atherosclerosis. Gut microbiota and its metabolites could affect the host immune system, which could also contribute to the development of atherosclerosis. It would be good to summarize the inflammation pathways in the development of atherosclerosis. The forth item, “the role of macrophages in atherosclerosis and reverse cholesterol transport”, the atherosclerosis pathology, is out of the scope of this review. It would be better to discuss the role of gut microbiota and its metabolites in the pathology of atherosclerosis. The authors mentioned (Lines 177-180) that most of polyphenols could not be absorbed directly by the small intestine and have to be converted to absorbable bioactive substances by gut microbiota. Please provide the bioactive substance of each polyphenols, including substrate, microbiota enzyme, and metabolite produce.

Author Response

"Please see the attachment ".

Round 2

Reviewer 2 Report

The manuscript is more improved. I have no further comments to mention.

Reviewer 3 Report

This manuscript has been improved. I have no additional concern.